# The Korean Medication Algorithm Project for Depressive Disorder (KMAP-DD): Changes in Preferred Treatment Strategies and Medications over 20 Years and Five Editions

**DOI:** 10.3390/jcm12031146

**Published:** 2023-02-01

**Authors:** Seung-Ho Jang, Won-Myong Bahk, Young Sup Woo, Jeong Seok Seo, Young-Min Park, Won Kim, Jong-Hyun Jeong, Se-Hoon Shim, Jung Goo Lee, Duk-In Jon, Kyung Joon Min

**Affiliations:** 1Department of Psychiatry, Wonkwang University Hospital, School of Medicine, Wonkwang University, Iksan 54538, Republic of Korea; 2Department of Psychiatry, College of Medicine, The Catholic University of Korea, Seoul 06591, Republic of Korea; 3Department of Psychiatry, College of Medicine, Chung-Ang University, Seoul 06974, Republic of Korea; 4Department of Psychiatry, Ilsan Paik Hospital, Inje University College of Medicine, Goyang 10380, Republic of Korea; 5Department of Psychiatry, Sanggye Paik Hospital, Inje University College of Medicine, Seoul 01757, Republic of Korea; 6Department of Psychiatry, Soonchunhyang University Cheonan Hospital, College of Medicine, Soonchunhyang University, Cheonan 31151, Republic of Korea; 7Department of Psychiatry, Haeundae Paik Hospital, College of Medicine, Inje University, Busan 48108, Republic of Korea; 8Department of Psychiatry, Hallym University Sacred Heart Hospital, Hallym University College of Medicine, Anyang 14068, Republic of Korea

**Keywords:** KAMP-DD, depressive disorder, expert consensus, treatment strategy

## Abstract

Background. The Korean Medication Algorithm Project for Depressive Disorder (KMAP-DD) is an expert consensus guideline for depressive disorder created in 2002, and since then, four revisions (2006, 2012, 2017, 2021) have been published. In this study, changes in the content of the KMAP-DD survey and recommendations for each period were examined. Methods. The development process of the KMAP-DD was composed of two stages. First, opinions from experts with abundant clinical experience were gathered through surveys. Next, a final guideline was prepared through discussion within the working committee regarding the suitability of the results with reference to recent clinical studies or other guidelines. Results. In mild depressive symptoms, antidepressant (AD) monotherapy was preferred, but when severe depression or when psychotic features were present, a combination of AD and atypical antipsychotics (AD + AAP) was preferred. AD monotherapy was preferred in most clinical subtypes. AD monotherapy was preferred for mild depressive symptoms, and AD + AAP was preferred for severe depression and depression with psychotic features in children, adolescents, and the elderly. Conclusions. This study identified the changes in the KMAP-DD treatment strategies and drug preferences in each period over the past 20 years. This work is expected to aid clinicians in establishing effective treatment strategies.

## 1. Introduction

With the development of psychopharmacology, new drugs have been introduced to treat depressive disorders. Evidence of the treatment effects and the switching or combining of drugs with different mechanisms, according to the characteristics of depressive symptoms, has constantly been accumulating. The choice of medication or treatment strategies in mental disorders, including major depressive disorder, has been traditionally determined by the clinician and considers the clinical characteristics of the patient. However, it is not easy for clinicians to keep pace with the rapid changes in psychopharmacology and incorporate them into their clinical practice. Therefore, providing guidelines on which treatment strategy to select to derive the best treatment outcome in clinical practice can be of great help to clinicians.

The guidelines largely consist of evidence-based and expert consensus guidelines [1]. An evidence-based guideline is formulated based on the results of a randomized controlled trial conducted while adhering to strict inclusion and exclusion criteria for subjects or the meta-analysis results of the randomized controlled trial. However, it is difficult to reflect the results of evidence-based guidelines in clinical practice even though the guidelines are highly likely to reach a clear and valid conclusion. Additionally, a meta-analysis presents problems such as a lack of study subjects, inadequate statistical verification power, different research designs, errors in the data extraction process, and differences in the assessment scales [2]. On the other hand, expert consensus guidelines ask experts for their opinions on effective treatments and reach an agreement based on those opinions. Although the reliability and validity of the survey results are questionable, the clinical reality is reflected [3]. Recently, attempts have been made to supplement the problems within each guideline by modifying the accepted guidelines that converge expert opinions and evidence-based treatments from meta-analyses or systemic reviews [4].

The Korean Medication Algorithm Project for Depressive Disorder (KMAP-DD) is an expert consensus guideline (KMAP-DD 2002) for major depressive disorder developed by the Korean Society for Affective Disorders in 2002 [5], and since then, four revisions (KMAP-DD 2006 [6], KMAP-DD 2012 [7], KMAP-DD 2017 [8], KMAP-DD 2021 [9,10] ,have been published updating the guidelines every 4–6 years. In this study, changes in the content of the KMAP-DD survey and recommendations for each period were examined. The accumulation of new evidence for the medication treatment of depressive disorders and the effects of drugs introduced into treatment strategies in clinical practices were investigated.

## 2. Development of the KMAP-DD

### 2.1. The Review Committee

The development process of the KMAP-DD was composed of two stages. First, opinions from experts with an abundance of clinical experience were gathered through surveys covering various clinical situations, and statistical processing was conducted to determine whether there was a consensus. Next, a final guideline was prepared through discussion within the working committee regarding the suitability of the results with reference to recent clinical studies or other guidelines. To this end, psychiatrists with abundant clinical experience in treating depressive disorders and high academic achievement were selected, and a survey was conducted by university hospitals, mental health medical hospitals, and clinics according to the type of treatment. A small reward was given to the experts who replied to the questionnaire. The response rates for the KMAP-DD questionnaire by period were 43/60, 66.15% (KMAP-DD 2002); 67/101, 66.3% (KMAP-DD 2006); 67/123, 54.5% (KMAP-DD 2012); 79/144, 54.9% (KMAP-DD 2017); and 98/143, 68.5% (KMAP-DD 2021).

### 2.2. The Questionnaire

The KMAP-DD questionnaire is based on the U.S. expert consensus guideline series [11], like the Korean Medication Algorithm Project for Bipolar Disorder [12] developed in South Korea. First developed in 2022 following APA guidelines, the KMAP-DD consists of the Texas Implementation of Medication Algorithm, ECG guidelines for the treatment of depression in women, an Expert Consensus Guideline series, pharmacotherapy of depressive disorder in older patients, and the Canadian network for mood and anxiety treatment [13,14,15,16,17]. The questionnaire has been modified and supplemented according to South Korea’s situation in each period.

### 2.3. Rating Scale

The revised nine-point scale of the RAND Corporation was used to evaluate the treatment strategies and suitability of medication selection according to the depressive symptoms, the duration of medication use, and the number of medication replacements that were directly described by the evaluators [12]. It was specified in the questionnaire to choose an ideal and desirable treatment rather than considering a clinical reality.

### 2.4. Statistical Analysis

The scores evaluated by the reviewers were divided into three categories (①–③, ④–⑥, ⑦~-), and a χ^2^-test was conducted to determine whether the distribution of respondents differed in the three categories. In addition, the 95% confidence interval (CI) of the average score of each selection item was calculated, and each score was categorized according to the threshold value (exceeding 6.5 -> first-line/preferred choice; 3.5 - under 6.5 -> second-line/alternate choice; and under 3.5 -> third-line/alternate choice). First-line items evaluated by more than 50% of the reviewers as 9 points were named “Treatment of choice (TOC).”

## 3. Changes in the KMAP-DD Survey Questions

### 3.1. The KMAP-DD 2002

The KMAP-DD 2002 consisted of three sections (Section 1: Major depressive disorder without psychotic features; Section 2: Major depressive disorder with psychotic features; Section 3: General tactics and strategies for major depressive disorder) with a total of 276 questions.

### 3.2. The KMAP-DD 2006

The KMAP-DD 2006 consisted of 54 individual situations for 22 clinical situations, including four sections (Section 1: Major depressive disorder without psychotic features; Section 2: Major depressive disorder with psychotic features; Section 3: Treatment of melancholic and atypical types of major depressive episodes; Section 4: Treatment of dysthymia and depressive disorder in women). In the KMAP-DD 2006, major depressive disorder without psychotic features was subdivided into mild, moderate, and severe. In addition, atypical features, intrinsic features, mild depressive disorder, dysthymic disorder, female features (premenstrual dysphoric disorder and postpartum depression), and treatment strategies according to adverse effects (sedation, anticholinergic, gastrointestinal, sexual, and cardiovascular) were added to determine the subtype of depression and drug selection according to the clinical situation of patients.

### 3.3. The KMAP-DD 2012

The KMAP-DD 2012 consisted of a total of 44 clinical situations, including seven sections (Section 1: Major depressive disorder without psychotic features; Section 2: Major depressive disorder with psychotic features; Section 3: Maintenance therapy; Section 4: The subtype of depression; Section 5: Child/adolescent, The elderly/female; Section 6: Safety, adverse effects, comorbid physical illnesses; Section 7: Non-pharmacological biological treatment). In the KMAP-DD 2012, major depressive disorder without psychotic features was subdivided into mild, moderate, and severe. Treatment strategies for clinical subtypes were accompanied by seasonal patterns, melancholia features, elderly, child/adolescent, and physical illnesses (weight gain, seizures, insomnia, serotonin syndrome, suicidal ideation, diabetes mellitus, thyroid-related, hepatobiliary, and nephrological disorders) were added.

### 3.4. The KMAP-DD 2017

The KMAP-DD 2017 consisted of a total of 44 clinical situations, including seven sections (Section 1: Major depressive disorder without psychotic features; Section 2: Major depressive disorder with psychotic features, Section 3: The subtype of depression; Section 4: Maintenance therapy; Section 5: Child/adolescent, The elderly/female; Section 6: Safety, adverse effects, comorbid physical illnesses; Section 7: Non-pharmacological biological treatment). In the KMAP-DD 2017, persistent depressive disorder, mixed features, anxious features, and disruptive mood dysregulation disorder (DMDD) were added by reflecting the Diagnostic and Statistical Manual of Mental Disorder-5 (DSM-5) developed in 2013 [18], and treatment strategies for pregnancy were investigated.

### 3.5. The KMAP-DD 2021

The KMAP-DD 2021 consisted of a total of 44 clinical situations, including seven sections (Section 1: Treatment strategies for major depressive disorder; Section 2: The subtype of depression; Section 3: Treatment-resistant depression; Section 4: Safety, adverse effects, comorbid physical illnesses; Section 5: Child/adolescent; Section 6: The elderly/female; Section 7: Non-pharmacological biological treatment). In the KMAP-DD 2021, the major categories according to psychotic features were integrated and composed of mild to moderate, severe without psychotic features, and severe with psychotic features. According to the classification of the American Academy of Child and Adolescent Psychiatry, child/adolescent was classified into child (<14 years) and adolescent (14–17 years) [19], and the definition of treatment-resistant depression (TRD) was investigated. For the elderly, the preference for psychostimulants was examined based on the report that the combined treatment of citalopram and methylphenidate had a significant effect compared to each combined placebo treatment of the two drugs [20]. In addition, Esketamine (nasal spray) was included in the drug selection item.

## 4. Changes in the KMAP-DD Treatment Strategy

(1)Changes in the adjustment period of treatment strategies for major depressive episodes began with the issue of KMAP-DD 2006. A questionnaire on the waiting period before changing the treatment strategy began to be conducted for cases of insufficient response to the initial antidepressant treatment. In the KMAP DD 2006, the waiting period was “3.3–6.1 weeks (no response to optimal dosage)” and “2.8–4.9 weeks (no response to maximal dosage).” In the KMAP-DD 2012, the waiting period was “3.2–7.4 weeks (partial response). It was “2.9–6.4 weeks (mild to moderate)” and “2.8–6.1 weeks (severe)” in the KMAP-DD 2017. In the KMAP-DD 2021, the waiting period was 2.2–4.3 weeks (no response) and 3.3–6.1 weeks (partial response) for mild to moderate major depressive episodes, and 1.9–3.6 weeks (no response) and 2.9–5.2 weeks (partial response) for severe major depressive episodes. Although it is difficult to directly compare the adjustment period of treatment strategies due to differences in the questionnaire according to the KMAP-DD date, it was found that the period was shortened over time: from 3.2–7.4 weeks (2012) to 2.9–6.4 weeks (2017) and further to 1.9–6.1 weeks (2021). As studies have shown that early improvement is an important factor in future treatment outcomes in the medication treatment of depressive disorders [21], clinicians are actively changing treatment drugs from the initial stage of treatment (Table 1).(2)Changes in treatment strategies and drug preferences.

### 4.1. Major Depressive Episodes

#### 4.1.1. Major Depressive Episodes without Psychotic Features

In the KMAP-DD 2002, antidepressant (AD) monotherapy was the first-line strategy. In the KMAP-DD 2006, AD monotherapy was the TOC for mild and moderate major depressive episodes and the first-line strategy for severe major depressive episodes. Since the KMAP-DD 2012, AD monotherapy has been the TOC for mild and moderate major depressive episodes. In the case of severe major depressive episodes, AD monotherapy was used in the KMAP-DD 2012, and after the KMAP-DD 2017, AD and typical antipsychotics (AAP) combination (AD + AAP) was added to the first-line strategy. In addition, since the 2010s, AD + AAP has been preferred in the clinical field as its effects on intractable depression have been reported in several clinical studies and through meta-analysis [22,23,24,25]. However, the preference for AD + AAP as an initial treatment differs from foreign clinical guidelines that recommend AD monotherapy [26,27,28]. Other clinical guidelines recommend AD monotherapy on the grounds that there is still a lack of firm evidence that combination with AAPs is superior to AD monotherapy [14] in terms of its stability, drug tolerance [29], and cost-benefit [30]. However, AD monotherapy is limited due to the initial low response rate (40–60%) [31,32] and remission rate (20–30%) [33,34]; additionally, it takes about 4–6 weeks to show sufficient effects [35]. AD + AAP is a treatment strategy that has been continuously studied to overcome these limitations, and in a recent meta-analysis, superior treatment effects were reported compared to AD monotherapy [36]. However, the studies were conducted on patients for whom AD monotherapy had failed, causing the results to be limited in application to direct initial treatment. Considering foreign reports that more than 50% of depression patients take more than two to three drugs [37], combination therapy, including AAP, seems to be preferable in clinical situations, even if the evidence is not yet sufficient (Table 2).

#### 4.1.2. Major Depressive Episodes with Psychotic Features

In the KMAP-DD 2002, the combination of selective serotonin reuptake inhibitors (SSRI) and antipsychotics was the first-line strategy, and since the KMAP-DD 2006, AD + AAP has been consistently the TOC. The superior effects of AD + AAP on depression with psychotic features have been consistently reported [38,39,40,41]. For medication, the high preference for Escitalopram and Aripiprazole has been prominent since the KMAP-DD 2017. The fact that Aripiprazole has indications as an additional treatment for major depressive disorder in South Korea and that Escitalopram was evaluated as highly efficient and safe in previous studies [42,43] may have influenced clinicians’ choice of medication (Table 2).

### 4.2. Changes in Treatment Strategies and Drug Preferences according to Subtype

#### 4.2.1. Atypical Features

Since the KMAP-DD 2006 was issued, AD monotherapy has been consistently adopted as the first-line strategy. The characteristics of atypical features are weight gain, hypersomnia, and hyperphagia [44]. Therefore, clinicians prefer AD monotherapy due to the high risk of deterioration of atypical symptoms when other drugs are added to ADs (Table 3).

#### 4.2.2. Seasonal Pattern

Since the KMAP-DD 2006 was issued, AD monotherapy has been the first-line strategy. In terms of drugs, Escitalopram and Bupropion have been highly preferred. Moreover, unlike other subtypes, Bupropion is highly preferred for the seasonal pattern of illness. The preventive effect of Bupropion was reported in a meta-analysis of 1042 patients with seasonal affective disorder [45] (Table 3).

#### 4.2.3. Melancholia (Intrinsic Features)

In the KMAP-DD 200, AD monotherapy was the first-line strategy for intrinsic features. Since the KMAP-DD 2012, the term has been changed to melancholia, and AD monotherapy became the first-line strategy. In terms of medication, Escitalopram and Venlafaxine have been favored. In the past, tricyclic antidepressants (TCA) tended to be preferred over SSRIs for melancholic features [46], but recently, no significant difference in the effects between the two drugs has been found [47]. In the case of TCA, its use in South Korea is continuously declining [48], and Venlafaxine, a serotonin-norepinephrine reuptake inhibitor, is still highly preferred (Table 3).

#### 4.2.4. Persistent Depressive Disorder (Dysthymic Disorder)

For dysthymic disorder, AD monotherapy was the TOC in the KMAP-DD 2006 and 2012. Even after the term was changed to persistent depressive disorder according to the DSM-5 criteria, AD monotherapy was the TOC in both the KMAP-DD 2017 and 2021. In the meta-analysis containing 56 RCTs for patients with persistent depressive disorder, most ADs showed significant effects, and adverse effects were the main reason for discontinuing the medication [49]. Therefore, rather than combining other drugs, clinicians prefer AD monotherapy to minimize adverse effects and increase therapeutic effects (Table 3).

#### 4.2.5. Mixed Features

For mixed features, AD + AAP and AD + mood stabilizers (MS) were the first-line strategies in the KMAP-DD 2017 and 2021. Based on reports that AAP combinations, such as Lurasidone [50] and Ziprasidone [51], are effective, combination therapy has recently been the mainstay [52], and this may have influenced the selection of treatment strategies (Table 3).

#### 4.2.6. Anxious Features

For anxious features, AD monotherapy and AD + AAP were the first-line strategies in the KMAP-DD 2017 and 2021 (Table 3). In the case of anxious depression, various methods have been tried to increase the therapeutic effects since treatment outcome is known to be poor [53]. Among the various methods, AAP combinations, such as Aripiprazole [54] or Quetiapine [55], are believed to be effective (Table 3).

### 4.3. Changes in Treatment Strategies and Drug Preferences in Children and Adolescents

#### 4.3.1. Mild to Moderate

AD monotherapy was the first-line strategy in the KMAP-DD 2012 and TOC in the KMAP-DD 2017. In the KMAP-DD 2021, AD monotherapy was the TOC for children and the first-line strategy for adolescents. For both children and adolescents, Escitalopram was the TOC as an early antidepressant. For childhood and adolescent depression, Escitalopram was found to have a superior effect on function and symptom improvement compared to a placebo [56] and showed a greater lead in the adolescent group [57]. This result is different from the Canadian Network for Mood and Anxiety Treatment (CANMAT), in which Fluoxetine was recommended as level 1, the highest basis. The stability and fewer side effects of Escitalopram are being considered by clinicians [58] (Table 4).

#### 4.3.2. Severe without Psychotic Features

AD monotherapy was added to the KMAP-DD 2012, and AD + AAP was added to the KMAP-DD 2017 as the first-line strategy. In the KMAP-DD 2021, AD monotherapy and AD + AAP were the first-line strategies for both children and adolescents; in terms of medication, Escitalopram (child) and Escitalopram and Fluoxetine (adolescent) were the TOC. In the case of children and adolescents, about 40–90% of severe depression is accompanied by one or more mental disorders [59], and the treatment outcome is poor [19]. It can be said that clinicians’ strategies to increase the effectiveness in the early stages of treatment resulted in AAP combinations (Table 4).

#### 4.3.3. Severe with Psychotic Feature

Since the KMAP-DD 2012, AD + AAP has been the TOC, and among the drugs, Escitalopram and Aripiprazole are preferred.

#### 4.3.4. Disruptive Mood Dysregulation Disorder

In the DMDD investigated from the KMAP-DD 2017, a consensus was not achieved (Table 4).

### 4.4. Changes in Treatment Strategies and Drug Preferences in the Elderly

#### 4.4.1. Mild to Moderate

Since the KMAP-DD 2012, AD monotherapy and Escitalopram have been the TOC. Since the elderly often take different drugs for various physical diseases [60], the use of a single drug is preferred in consideration of drug interactions (Table 5).

#### 4.4.2. Severe without Psychotic Features

AD monotherapy was added as the first-line strategy in the KMAP-DD 2012, and AD + AAP has been the first-line strategy since the KMAP-DD 2017. In South Korea, the elderly often show severe depression due to long-neglected symptoms, and thus, the suicide risk for elderly people is very high [61]. Therefore, despite the adverse effects of AAP use for depression in the elderly [62], clinicians prefer the effective and quick results of adding AAP (Table 5).

#### 4.4.3. Severe with Psychotic Features

Since the KMAP-DD 2012, AD + AAP, Escitalopram, and Aripiprazole have been the TOC. In the case of Aripiprazole, it has been reported that combination with AD in elderly TRD shows high therapeutic effects [63] and fewer adverse effects, such as anticholinergic effects, somnolence, and metabolic syndrome [64], compared to other AAPs (Table 5).

### 4.5. Changes in Treatment Strategies and Drug Preferences in Females

#### 4.5.1. Premenstrual Dysphoric Disorder

AD monotherapy was the first-line strategy in the KMAP-DD 2006 and was selected as the TOC for the KMAP-DD 2017 (Table 6).

#### 4.5.2. Postpartum Depression

For mild to moderate depression, AD monotherapy was the first-line strategy in the KMAP-DD 2006 and TOC in the KMAP-DD 2017. AD + AAP was also added to the first-line strategy in the KMAP-DD 2021. For severe depression without psychotic features, AD monotherapy was the first-line strategy in the KMAP-DD 2006, and AD + AAP was added as the first-line strategy when the KMAP-DD 2012 was issued. For severe depression with psychotic features, AD + AAP has been the TOC since the KMAP-DD 2006, and MS + AAP was added as the first-line strategy only in the KMAP-DD 2012. There is a high risk of bipolar disorder when there is severe postpartum depression [65], and the recent treatment strategy of using AD + AAP from the onset of treatment of major depressive disorders [66,67] is reflected in Table 6.

#### 4.5.3. Pregnancy

In the case of mild to moderate depression, AD monotherapy was the TOC in the KMAP-DD 2017 but the first-line strategy in the KMAP-DD 2021. For severe depression without psychotic features, AD monotherapy was the first-line strategy. For severe depression with psychotic features, AD + AAP was the TOC in the KMAP-DD 2017, but there was a higher preference for elective convulsive therapy (ECT) in the KMAP-DD 2021 (Table 4). Severe depression that occurs during pregnancy is a psychiatric emergency with a high suicide risk [68]. Therefore, clinicians are attempting more active therapeutic interventions, and treatments with quick therapeutic effects like ECT are preferred in the process [69] (Table 6).

## 5. Definition of Treatment-Resistant Depression (TRD)

In the KMAP-DD 2021, a survey was conducted on the definition of TRD, where 21% of respondents answered, “inadequate response despite appropriate treatment using two or more antidepressants of different lines”, and 44% of respondents answered, “inadequate response despite appropriate treatment using two types of antidepressants and one atypical antipsychotic drug.” However, a consensus was not achieved. Even in a study by Gaynes et al. that analyzed 260 articles, a consensus on the definition of TRD could not be established; instead, “a minimum of two prior treatment failures and confirmation of prior adequate dose and duration” was suggested as the most common definition of TRD [70]. Therefore, further research is needed to establish a consensus on the definition of TRD in the future.

## 6. Conclusions

The KMAP-DD is a representative expert consensus guideline that investigates the treatment strategies of depressive disorders through questionnaires and has been revised four times in 2006, 2012, 2017, and 2021. Through the changes in the KMAP-DD at each period, this study investigated the effect of the emergence of new drugs or treatment methods on clinicians’ treatment strategies in the field.

In terms of the changes in treatment strategies shown in the KMAP-DD, the adjustment period to a new treatment strategy when patients showed insufficient response to the initially selected antidepressant monotherapy tended to shorten. In the case of mild depressive symptoms, AD monotherapy was preferred, but when accompanied by severe depression or psychotic features, AD + AAP was preferred. In South Korea, non-specialists in mental health, such as family members, friends, and doctors of Korean medicine, are perceived to help treat depression, and psychiatric treatment is avoided due to fear of stigma [71]. Consequently, proper treatment is often not received in the early stages of depression but is often initiated during a state of severe depression. Therefore, clinicians prefer a quick and effective treatment strategy and quickly change the treatment strategy if the treatment effect is insufficient.

In the case of treatment strategies by subtype, AD monotherapy was preferred in most clinical subtypes, but there was a difference in the drug selection of each subtype. Similar to the adult group, AD monotherapy was preferred for mild cases, and AD + AAP was preferred for severe depression and depression with psychotic features in vulnerable age groups, such as children, adolescents, and the elderly. However, the side effects of the medication were considered more significantly when choosing a treatment strategy for those age groups. In the case of females, unlike in the past, AAP or ECT was actively considered to prevent psychiatric emergencies, such as bipolar disorder or suicide.

There are some limitations in this study. First, in the production of an algorithm, the opinions of experts and the latest medication treatment trends were reflected rather than in reference to objectively verified data. Therefore, some reviewers may have different views depending on the question. Secondly, even if the experts reach a consensus, it may not be suitable for clinical situations. Third, it is regrettable that the contents of psychotherapy and social and environmental approaches were omitted. Fourth, Ketamine (nasal spray) was included in the KMAP-DD 2021 questionnaire but not in the medication algorithm itself because only a few experts had experience with the product. As the use of Ketamine (nasal spray and intravenous) is increasing in South Korea, it is expected to be included in the revised KMAP-DD. Fifth, the KMAP-DD is a medication-centered system; thus, other neurostimulation therapies were not included, except for frequently used ECTs.

Despite these limitations, this study identified the changes in the KMAP-DD treatment strategies and drug preferences in each period over the past 20 years and investigated the effects of introducing new drugs and treatment methods on clinicians’ treatment strategies. This is expected to aid clinicians in establishing effective treatment strategies in the clinical field.

## Figures and Tables

**Table 1 jcm-12-01146-t001:** Changes in the adjustment period of treatment strategies in KMAP-DD.

	2002	2006		2012		2017		2021	
Duration of medication maintenance (Week)	-	No response to optimal dosage	3.3–6.1	Partial response to AD monotherapy	3.2–7.4	Mild to moderate	2.9–6.4	No response	2.2–4.3
		No response to maximal dosage	2.8–4.9					Partial response	3.3–6.1
						Severe	2.8–6.1	No response	1.9–3.6
								Partial response	2.9–5.2

KMAP-DD: Korean Medication Algorithm Project for Depressive Disorder, AD: Antidepressant.

**Table 2 jcm-12-01146-t002:** First-line treatment strategies for major depressive episode of KMAP-DD.

	2002		2006			2012			2017		2021		
Without psychotic feature	AD	*Paroxetine fluoxetine sertraline*	Mild	AD *	*SSRI Venlafaxine*	Mild to moderate	AD *	*Escitalopram Sertraline* *Paroxetine*	AD *	*Escitalopram * Sertraline * Fluoxetine*	AD *		*Escitalopram ** *Sertraline* *Desvenlafaxine*
			Moderate	AD *	*SSRI, Venlafaxine Mirtazapine*								
			Severe	AD	*Venlafaxine* *SSRI* *Mirtazapine*	Severe	AD	*Venlafaxine Mirtazapine Escitalopram*	AD AD + AAP	*Escitalopram ** *Venlafaxine ** *Mirtazapine ** *Aripiprazole ** *Quetiapine*	AD AD + AAP		*Escitalopram * Desvenlafaxine Venlafaxine* *Aripiprazole*
With psychotic feature	SSRI + AAP		Severe with psychotic feature	AD + AAP *	*Venlafaxine SSRI Mirtazapine* *Quetiapine Risperidone*	With psychotic feature	AD + AAP *	*Mirtazapine Escitalopram Venlafaxine* *Quetiapine* *Aripiprazole Olanzapine*	AD + AAP *	*Escitalopram ** *Aripiprazole **	Severe with psychotic feature	AD + AAP *	*Escitalopram * Venlafaxine Desvenlafaxine Aripiprazole ** *Quetiapine Olanzapine*

* Treatment of choice, KMAP-DD: Korean Medication Algorithm Project for Depressive Disorder, AD: Antidepressant, AAP: Atypical antipsychotics, SSRI: selective serotonin reuptake inhibitor.

**Table 3 jcm-12-01146-t003:** First-line treatment strategies for the subtypes of depression of KMAP-DD.

	2006		2012			2017			2021	
Atypical feature	AD	*SSRI* *Venlafaxine*	AD		*Escitalopram Fluoxetine* *Sertraline*	AD	*Escitalopram,* *fluoxetine,* *Sertraline*		AD	*Escitalopram* *Desvenlafaxine* *Fluoxetine*
Intrinsic feature	AD	*SSRI* *Venlafaxine* *Mirtazapine*	Melancholia	AD	*Escitalopram Fluoxetine* *Sertraline*	AD	*Escitalopram ** *Venlafaxine **		AD	*Escitalopram ** *Desvenlafaxine* *Venlafaxine*
Mild depressive disorder	AD *	*SSRI*	Seasonal pattern	AD	*Escitalopram Fluoxetine* *Sertraline*	AD	*Escitalopram* *fluoxetine* *Paroxetine*		AD	*Escitalopram* *Sertraline* *fluoxetine*
Dysthymic disorder	AD *	*SSRI*	AD *	*Escitalopram Fluoxetine* *Sertraline*	Persistent depressive disorder	AD *	*Escitalopram* *Fluoxetine* *Paroxetine*		AD *	Escitalopram * Desvenlafaxine Sertraline
						Mixed feature	AD + AAP AD + MS		AD + AAP AD + MS	*Escitalopram* *Sertraline* *Fluoxetine* *Aripiprazole* *Quetiapine* *Olanzapine*
						Anxious feature	AD + AAP AD		AD AD + AAP	*Escitalopram* *Fluoxetine* *Paroxetine* *Aripiprazole* *Quetiapine*

* Treatment of choice, KMAP-DD: Korean Medication Algorithm Project for Depressive Disorder, AD: Antidepressant, AAP: Atypical antipsychotics, MS: Mood stabilizer.

**Table 4 jcm-12-01146-t004:** First-line-treatment strategies for children and adolescents of KMAP-DD.

	2012	2017		2021		
Mild to moderate depression	AD	*AD**	*Escitalopram* *Fluoxetine*	Child (<14 yrs)	AD *	*Escitalopram ** *Fluoxetine*
				Adolescent (14–19 yrs)	AD	*Escitalopram ** *Fluoxetine* *Sertraline*
Severe depression without psychotic feature	AD	AD AD + AAP	*Escitalopram* *Fluoxetine*	Child (<14 yrs)	AD, AD + AAP	*Escitalopram ** *Fluoxetine* *Sertraline*
				Adolescent (14–19 yrs)	AD, AD + AAP	*Fluoxetine ** *Escitalopram ** *Sertraline*
Severe depression with psychotic feature	AD + AAP *	AD + AAP *	*Escitalopram* *Fluoxetine* *Aripiprazole* *Risperidone*	Child (<14 yrs)	AD + AAP *	*Escitalopram* *Fluoxetine* *Sertraline* *Aripiprazole ** *Risperidone* *Quetiapine*
				Adolescent (14–19 yrs)	AD + AAP *	*Escitalopram ** *Fluoxetine* *Sertraline* *Aripiprazole ** *Risperidone*
		Disruptive mood dysregulation disorder	Non-Consensus	Non-Consensus		

* Treatment of choice, KMAP-DD: Korean Medication Algorithm Project for Depressive Disorder, AD: Antidepressant, AAP: Atypical antipsychotics.

**Table 5 jcm-12-01146-t005:** First-line treatment strategies for elderly of KMAP-DD.

2012			2017		2021	
Mild to moderate depression	AD *	*Escitalopram **	AD *	*Escitalopram ** *Sertraline* *Duloxetine*	AD *	*Escitalopram ** *Sertraline* *Desvenlafaxine*
Severe depression without psychotic feature	AD		AD AD + AAP	*Escitalopram ** *Sertraline* *Duloxetine*	AD AD + AAP	*Escitalopram ** *Sertraline* *Desvenlafaxine*
Severe depression with psychotic feature	AD + AAP *		AD + AAP *	*Escitalopram ** *Fluoxetine* *Sertraline* *Aripiprazole ** *Quetiapine*	AD + AAP *	*Escitalopram ** *Sertraline* *Desvenlafaxine* *Aripiprazole ** *Quetiapine*

* Treatment of choice, KMAP-DD: Korean Medication Algorithm Project for Depressive Disorder, AD: Antidepressant, AAP: Atypical antipsychotics.

**Table 6 jcm-12-01146-t006:** First and second-line treatment strategies for females of KMAP-DD.

	2006		2012		2017	2021			
Premenstrual dysphoric disorder	AD		AD	*Escitalopram* *Fluoxetine* *Sertraline*	AD *	*Escitalopram * Fluoxetine Sertraline*		AD *	*Fluoxetine* *Escitalopram* *Sertraline*
Postpartum depression	Mild to moderate	AD	AD		AD *			AD, AD + AAP	
	Severe without psychotic feature	AD	AD AD + AAP		AD AD + AAP			AD + AAP	
	Severe with psychotic feature	*AD + AAP **	AD + AAP * MS + AAP		AD + AAP *			AD + AAP *	
					Pregnancy	Mild to moderate	AD *	AD	
						Severe without psychotic feature	AD	AD	
						Severe with psychotic feature	AD + AAP * ECT	ECT AD + AAP	

* Treatment of choice, KMAP-DD: Korean Medication Algorithm Project for Depressive Disorder, AD: Antidepressant, AAP: Atypical antipsychotics. MS: Mood stabilizer, ECT: Electroconvulsive therapy.

## Data Availability

The datasets generated and/or analyzed during the current study are available from the corresponding author upon reasonable request.

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
