# Peer review of "The Korean Medication Algorithm Project for Depressive Disorder (KMAP-DD): Changes in Preferred Treatment Strategies and Medications over 20 Years and Five Editions"

_jcm, 2023, doi:10.3390/jcm12031146_

Round 1
Reviewer 1 Report
I appreciate the opportunity to review this paper, containing the revised version of the The Korean Medication Algorithm Project for Depressive Disorder (KMAP-DD). The paper is well written and, similarly to its previous versions, offers an important contribution to clinicians. I understand that, as an expert consensus paper, there are restrictions in terms of revisions and additions. Nonetheless, but I would like to mention a few points/reservations:
1) I missed a more clear position of the algorithm with regards to ketamine treatment. The authors state that esketamine was included among the options, but I was not able to find it among the recommendations. I assume it was ranked low by the experts who participated in the consensus. There is no mention to intravenous ketamine (I assume due to the same reason). I would suggest at least a brief mention to it in the discussion (meaning, some kind of explanation regarding the absence of ketamine/esketamine among the recommended strategies, if the authors feel like that would be appropriate.
2) I have the same concerns regarding Brexanolone for pospartum depression
3) Neurostimulation therapies: the algorithm does include ECT but does not mention other forms of stimulation (TMS, VNS, etc)
4) In the introduction, the following sentence seems to be odd, I believe there was a mistake in the year mentioned: "The Korean Medication Algorithm Project for Depressive Disorder (KMAP-DD) is an expert consensus guideline (KMAP-DD 2002) for major depressive disorder developed by the Korean Society for Affective Disorders in 2022, and since then, four revisions (KMAP- DD 2006, KMAP-DD 2012, KMAP-DD 2017, KMAP-DD 2021) have been published every 4-6 years." I assume the authors meant "... developed by the Korean Society for Affective Disorders in 2002", not in 2022.
Author Response
1) I missed a more clear position of the algorithm with regards to ketamine treatment. The authors state that esketamine was included among the options, but I was not able to find it among the recommendations. I assume it was ranked low by the experts who participated in the consensus. There is no mention to intravenous ketamine (I assume due to the same reason). I would suggest at least a brief mention to it in the discussion (meaning, some kind of explanation regarding the absence of ketamine/esketamine among the recommended strategies, if the authors feel like that would be appropriate.
: Ketamine (nasal spray) was included in the 2021 Korean Medication Algorithm for Depressive Disorder (KMAP-DD) questionnaire, but not in the medication algorithm itself because only a few experts had experience with the product. As the use of ketamine (nasal spray and intravenous) is increasing in South Korea, it is expected to be included in the revised KMAP-DD.
- We added this content in discussion.
2) I have the same concerns regarding Brexanolone for pospartum depression
: The use of brexanolone is not approved in South Korea. Thus, it was not included in the KMAP-DD survey items.
3) Neurostimulation therapies: the algorithm does include ECT but does not mention other forms of stimulation (TMS, VNS, etc)
: Because the KMAP-DD is a medication-centered system, other neurostimulation therapies and psychotherapy were not included, except for frequently used ECTs.
- We added this content in discussion.
4) In the introduction, the following sentence seems to be odd, I believe there was a mistake in the year mentioned: "The Korean Medication Algorithm Project for Depressive Disorder (KMAP-DD) is an expert consensus guideline (KMAP-DD 2002) for major depressive disorder developed by the Korean Society for Affective Disorders in 2022, and since then, four revisions (KMAP- DD 2006, KMAP-DD 2012, KMAP-DD 2017, KMAP-DD 2021) have been published every 4-6 years." I assume the authors meant "... developed by the Korean Society for Affective Disorders in 2002", not in 2022.
- We revised this sentence according to the reviewer’s comment.
Reviewer 2 Report
This is a qualitative analysis of the difference in the four iterations of the KMAP-DD, an algorithmic approach to the treatment of Depression developed and implemented in South Korea. The article is well-written and offers a unique historical perspective on the progression of clinical expertise coupled with the scientific literature. The paper does not appear to list its own methods and its significance is somewhat stymied by its historical emphasis.
1.) In the abstract methods, you discuss the development of the scale and not your own process on how you went about reviewing the scales and noticing the differences. You should list your own methods here and describe them in depth in the article, otherwise the article looks more like a historical record.
2.) In addition to the above, there is a missed opportunity here to critique the survey and possibly influence future iterations. You touched upon psychotherapy and wrap-around services not being in the survey. You should expand. What about ECT + antidepressant + antipsychotics or any combination thereof for pregnancy? or treatment in depression with catatonic features? By discussing some of the other gaps, you may increase the significance of your content and have a larger impact.
3.) Table 1 is introduced to early and doesn't go with the material referencing it on page 2.
4.) The dates for KMAP-DD are mixed up throughout the manuscript, impeding readability. Double-check that you are referencing the right version and year of origin.
5.) I'm confused by the sentence:
"Recently, attempts have been made to counteract the problems of each guideline by compromising the evidence..." Is 'compromising' the correct word here?
The sentence in the conclusion, "...was not pursued received earlier." should be revised.
6.) In the seasonal affective section you mention a study with 15 patients reporting superior effects of Buproprion. This is likely inadequately powered and seems somewhat comical when combined with the massive meta-analysis in the same sentence. Consider removing or adding more references if available to make your point.
Author Response
1.) In the abstract methods, you discuss the development of the scale and not your own process on how you went about reviewing the scales and noticing the differences. You should list your own methods here and describe them in depth in the article, otherwise the article looks more like a historical record.
: This study aimed to elucidate the changes in the past 20 years, rather than to summarize the contents of the KMAP-DD at a specific time. In addition, due to the 200-word restriction on the abstract as per the submission rules, the scale was described in detail in the main text.
2.) In addition to the above, there is a missed opportunity here to critique the survey and possibly influence future iterations. You touched upon psychotherapy and wrap-around services not being in the survey. You should expand. What about ECT + antidepressant + antipsychotics or any combination thereof for pregnancy? or treatment in depression with catatonic features? By discussing some of the other gaps, you may increase the significance of your content and have a larger impact.
: Because the KMAP-DD is a medication-centered system, other neurostimulation therapies and psychotherapy were not included, except for frequently used ECTs.
-We added this content in discussion.
3.) Table 1 is introduced to early and doesn't go with the material referencing it on page 2.
-We will request the editorial team to adjust the position of Table 1.
4.) The dates for KMAP-DD are mixed up throughout the manuscript, impeding readability. Double-check that you are referencing the right version and year of origin.
-The version and year of KMAP-DD were double-checked per the reviewer's comment.
5.) I'm confused by the sentence:
"Recently, attempts have been made to counteract the problems of each guideline by compromising the evidence..." Is 'compromising' the correct word here?
-We revised the content according to the reviewer’s comments.
“Recently, attempts are being made to supplement the problems of each guideline by modifying the agreed guidelines that converged expert opinions and evidence-based treatment guidelines from meta-analyses or systemic reviews.”
The sentence in the conclusion, "...was not pursued received earlier." should be revised.
-We revised the content according to the reviewer’s comments.
“Consequently, proper treatment is often not received in the early stages and is often initiated during the state of severe depression.”
6.) In the seasonal affective section you mention a study with 15 patients reporting superior effects of Buproprion. This is likely inadequately powered and seems somewhat comical when combined with the massive meta-analysis in the same sentence. Consider removing or adding more references if available to make your point.
- The content of reference 45 has been deleted as per the reviewer's comment.
“Preventive effect of Bupropion was reported in a meta-analysis of 1,042 patients with seasonal affective disorder.45”